# Proton Bragg Peak FLASH Enables Organ Sparing and Ultra-High Dose-Rate Delivery: Proof of Principle in Recurrent Head and Neck Cancer

**DOI:** 10.3390/cancers15153828

**Published:** 2023-07-28

**Authors:** Michael Pennock, Shouyi Wei, Chingyun Cheng, Haibo Lin, Shaakir Hasan, Arpit M. Chhabra, J. Isabelle Choi, Richard L. Bakst, Rafi Kabarriti, Charles B. Simone II, Nancy Y. Lee, Minglei Kang, Robert H. Press

**Affiliations:** 1Department of Radiation Oncology, Albert Einstein College of Medicine, Montefiore Medical Center, New York, NY 10461, USA; rkabarri@montefiore.org; 2Department of Physics, New York Proton Center, New York, NY 10035, USA; alexwei8878@gmail.com (S.W.); hlin@nyproton.com (H.L.); shasan@nyproton.com (S.H.); mkang@nyproton.com (M.K.); 3Department of Radiation Oncology, Rutgers Cancer Institute of New Jersey, New Brunswick, NJ 08901, USA; chingyun.cheng@rutgers.edu; 4Department of Radiation Oncology, New York Proton Center, New York, NY 10035, USA; achhabra@nyproton.com (A.M.C.); ichoi@nyproton.com (J.I.C.); csimone@nyproton.com (C.B.S.II); 5Department of Radiation Oncology—Radiation Oncology Associates, Icahn School of Medicine at Mount Sinai, New York, NY 10029, USA; richard.bakst@mountsinai.org; 6Department of Radiation Oncology, Memorial Sloan Kettering Cancer Center, New York, NY 10065, USA; leen2@mskcc.org; 7Department of Radiation Oncology, Baptist Health South Florida, Miami Cancer Institute, Miami, FL 33176, USA; robert.press@baptisthealth.net

**Keywords:** proton, radiation therapy, dosimetry, toxicity, pencil beam, planning, safety

## Abstract

**Simple Summary:**

Bragg peak proton FLASH is a novel way to deliver radiotherapy to cancer that combines ultra-high dose rates and conformal organ sparing provided by the physical characteristics of protons. Increasing the dose rate to ultra-high rates may decrease normal tissue toxicity through unique biological mechanisms that allow normal organs to be preserved. Most investigations of FLASH radiotherapy have used transmission beams that traverse the entire patient in order to generate high enough dose rates. We were able to create Bragg peak FLASH by making novel modifications to the planning software and the treatment delivery of the machine. This study demonstrates that conventional dose rate proton therapy and proton Bragg peak FLASH display similar organ sparing when plans were created and compared within the same patient. Bragg peak proton FLASH improves organ sparing over transmission proton FLASH, while both modalities have similar target and FLASH dose-rate coverage. Thus, Bragg peak proton FLASH may improve the therapeutic ratio in head and neck reirradiation.

**Abstract:**

Proton pencil-beam scanning (PBS) Bragg peak FLASH combines ultra-high dose rate delivery and organ-at-risk (OAR) sparing. This proof-of-principle study compared dosimetry and dose rate coverage between PBS Bragg peak FLASH and PBS transmission FLASH in head and neck reirradiation. PBS Bragg peak FLASH plans were created via the highest beam single energy, range shifter, and range compensator, and were compared to PBS transmission FLASH plans for 6 GyE/fraction and 10 GyE/fraction in eight recurrent head and neck patients originally treated with quad shot reirradiation (14.8/3.7 CGE). The 6 GyE/fraction and 10 GyE/fraction plans were also created using conventional-rate intensity-modulated proton therapy techniques. PBS Bragg peak FLASH, PBS transmission FLASH, and conventional plans were compared for OAR sparing, FLASH dose rate coverage, and target coverage. All FLASH OAR V40 Gy/s dose rate coverage was 90–100% at 6 GyE and 10 GyE for both FLASH modalities. PBS Bragg peak FLASH generated dose volume histograms (DVHs) like those of conventional therapy and demonstrated improved OAR dose sparing over PBS transmission FLASH. All the modalities had similar CTV coverage. PBS Bragg peak FLASH can deliver conformal, ultra-high dose rate FLASH with a two-millisecond delivery of the minimum MU per spot. PBS Bragg peak FLASH demonstrated similar dose rate coverage to PBS transmission FLASH with improved OAR dose-sparing, which was more pronounced in the 10 GyE/fraction than in the 6 GyE/fraction. This feasibility study generates hypotheses for the benefits of FLASH in head and neck reirradiation and developing biological models.

## 1. Introduction

Concurrent chemoradiotherapy (CRT) or surgery followed by radiotherapy (RT) with or without chemotherapy are important treatment options in non-human papillomavirus-related (HPV) head and neck cancer, which has a global incidence of 690,000 cases and a five-year survival of approximately 50% [1,2]. However, locoregional disease recurrence is not uncommon and often causes morbidity and death [3]. Salvage reirradiation can be associated with substantial toxicity and morbidity [4]. Radiation doses to organs at risk (OAR) are dose-limiting, and treatment efficacy may decrease if OAR preservation is prioritized over tumor coverage. Despite the evolution of photon intensity-modulated RT (IMRT), undesirable doses are still deposited in critical OARs. Advances in pencil-beam scanning (PBS) proton therapy depositing dose at the Bragg peak, with little exit dose to distal OARs, have enhanced the conformality and therapeutic ratio in some scenarios, maintaining comparable control to IMRT with less toxicity [5,6,7,8,9]. PBS allows for greater benefits than passive scattering via heterogenous dose distribution within each beam [10,11,12].

Due to prior receipt of high-dose radiation, head and neck reirradiation can cause numerous acute and late complications, and there remains a great need for an improved therapeutic ratio [13]. Radiation dose is a risk factor for carotid intimal disease leading to rupture and stroke [14,15], and causes temporal lobe injury (TLI), temporal lobe necrosis (TLN) [16], and cognitive impairment in a volume-dependent fashion [17,18,19]. Proton therapy is often considered in this context for conformality and minimization of cumulative dose overlap [20,21,22].

FLASH-RT (ultra-high dose rate > 40 Gy/s) is a groundbreaking modality that has demonstrated superior normal tissue sparing and similar local control to conventional-rate radiotherapy modalities in pre-clinical models [23,24,25]. Murine studies have demonstrated tissue sparing effects at the beam entrance and Bragg peak regions for proton FLASH [26,27], and decreased toxicity has been associated with the shortest delivery time and pulse number [28,29]. FLASH-RT has also been shown to reduce fibrosis, inflammation, and senescence [23,24,30]. Research is ongoing to elucidate the underlying FLASH radiobiology. Theories include oxygen depletion or reactive oxygen species prevention [31,32] and variations in DNA damage [33,34]. FLASH has demonstrated tumor control and minimal normal tissue damage in a human case [35]. This potential increase in the therapeutic ratio and conformality offered by FLASH has generated exploration into FLASH delivery via modifying existing systems [36], optimizing beam and field properties [36,37], creating new hardware and software [37], and developing new biological models [38].

Original studies of proton FLASH, including the first FLASH clinical trial in the setting of osseous limb metastases [39,40], have necessitated transmission beams to achieve ultra-high dose rates, which traverse the entire patient and do not spare critical structures [41,42,43]. It is still desirable to reduce the dose as much as possible in normal tissues, even if FLASH can be delivered. PBS achieves conformity by steering beamlets with magnets, and can deliver ultra-high nozzle currents [44,45]. Achieving FLASH is complicated by monitor unit (MU) and dose-rate quantification, high energy requirements, spot and layer dynamics, and other factors [41,44,46,47,48,49]. While PBS Bragg peak FLASH has almost zero exit dose in theory, the extent of its dosimetric performance remains undefined. By combining modifications to the range shifter, range compensator, spot map optimization, and inverse planning system, we propose that PBS Bragg peak FLASH is capable of conformality via Bragg peak, superior OAR sparing to PBS transmission FLASH, and adequate FLASH dose rate coverage [42].

This proof-of-principle work evaluated the dosimetric potential of the novel PBS Bragg peak FLASH and compared it to that of PBS transmission FLASH and conventional-rate intensity-modulated proton therapy in a cohort of reirradiation head and neck patients. The FLASH treatment plans were evaluated for OAR sparing and the achievement of sufficiently high FLASH dose rate coverage with realistic dose thresholds under multiple field optimization. We predicted that Bragg peak FLASH would yield superior plan quality while still achieving sufficient FLASH dose-rate coverage for head and neck reirradiation.

## 2. Materials and Methods

The New York Proton Center (NYPC) physics team developed novel planning software based on the matRad (German Cancer Research Center DKFZ, Heidelberg, Germany) framework [50] that utilizes the universal range shifter and beam-specific range compensator to pull back the proton range of the cyclotron’s highest single-energy beam to generate intensity-modulated proton therapy (IMPT)-equivalent FLASH treatment plans. By eliminating the exit dose, the Bragg peak FLASH plans were expected to have better OAR sparing. The dose was reported in “GyE” units to account for a proton RBE of 1.1, and to show that RT was delivered in bio-equivalent doses to well-known photon RT prescriptions and OAR constraints. Similar target coverage goals and dose constraints were utilized during optimization to facilitate equal comparisons between the treatment methods. This method has been previously described [42,43,51].

The Varian ProBeam (Varian Medical Systems, Palo Alto, CA, USA) models were configured using an in-house platform with a 2-millisecond (ms) delivery time (MST) of the minimum monitor units (MU) per spot [48], 250 megavoltage (MeV) beams, and a 10 mm (mm) per ms spot scanning speed under realistic FLASH settings [44]. In this system, the minimal spot duration in layer-wise delivery was 2 ms or larger; thus, there was a minimum MU requirement in FLASH mode, and the higher beam current used for FLASH required a larger minimal MU per spot to achieve the 2-ms threshold. To stay within these parameters, we set the minimum MU to 300–500 per spot and achieved a 2-ms MST, generating currents of 125–215 nA and facilitating less-hot dose distributions than planning for higher MUs per spot [48,51,52]. This system’s maximal beam current at isocenter was 215 nA with this configuration, was stable in FLASH mode, and applied to all PBS spots in the field. A sharpened penumbra is always critical to achieve high-quality treatment for head and neck cancers. Even though the system generated a small spot size of 2.5 mm for 250 MeV via quadrupole magnets, we still considered the impact of enlarged spot size from the scattering effect from the range shifter and range compensator. Thus, we placed these range-pullback devices as close as possible to the patient during the treatment planning design to minimize the lateral penumbra without the use of an aperture. This aspect will be investigated in future studies.

We quantified the dose rates using previously described methods to generate a dose-averaged dose rate (DADR) and introduced plan volume covered by 40 Gy/s (V40 Gy/s) to quantify CTV and OAR FLASH dose-rate coverage [41,48]. A spot-map optimization tool accounted for dose rate, spot times, plan quality, and minimum MU constraints in inverse planning, as previously described, and allowed for conformal, IMPT-equivalent dosimetry via inverse multiple-field optimization [51]. OAR sparing and target coverage were evaluated via dose–volume histograms (DVH). Dose-rate volume histograms (DRVH) were compared between FLASH modalities.

After IRB approval (IRB: 20201708, approved 19 June 2020), PBS Bragg peak and transmission FLASH plans (≥40 Gy/s) were created, optimized, and compared for 8 recurrent head and neck patients previously treated with conventional-rate IMPT “quad shot” RT (14.8 GyE in 4 3.7-GyE fractions, 2 per day). In line with research showing that FLASH tissue sparing starts at 40 Gy/s and 5–10 Gy per fraction [52,53], as well as the need to assess clinically relevant doses that are feasible with current technology, 6-GyE single-fraction and 10-GyE single-fraction plans were optimized and compared across conventional rate IMPT, PBS Bragg peak FLASH (≥40 Gy/s), and PBS transmission FLASH (≥40 Gy/s), resulting in 6 plans per patient. All CTVs were GTV expansions.

The Bragg peak FLASH method only uses the single highest-energy 250-MeV beam from the cyclotron to achieve ultra-high FLASH dose rates. When the spot scanning time was incorporated into the dose rate calculation, there was a maximal field size based on the maximal beam current. According to our commissioning results, the maximal field size was approximately 8 × 8 cm^2^ under our maximum current of 215 nA. As the dose rate in the near-Bragg peak region can be ~40% lower compared to the plateau region and worsened by air gaps, we overcame this with the aforementioned high minimum MU per spot and high nozzle current [48]. There was no spread out Bragg peak (SOBP) in any single beam direction in FLASH mode when using single energy beams and modulating them with a range shifter (energy straggling) and beam-specific compensator at the end of the nozzle. Multiple beams from different angles were used generate a uniform region. A smaller air gap of 5 cm was used for treatment plan optimization and reducing the penumbra. A 3-cm range shifter was used, as is typical of head and neck plans.

FLASH plans were generated using the same beam arrangement as the conventional-rate plans to provide optimal quality, ultra-high dose rates, uniform dose distribution, and minimal bias in plan comparisons [42]. Well-separated beams are favorable in transmission FLASH for preventing an increased dose beyond the target, which is avoided in Bragg peak FLASH [54].

Beam 3D dose rates were computed based on DADR. DRVHs for CTV and OARs were calculated to evaluate FLASH dose-rate (>40 Gy/s) coverage. DADR was defined for each voxel as the dose-weighted mean of the instantaneous dose rates of all spots, as previously described [41].

The target coverage was normalized to 100% CTV, receiving at least 95% of the prescribed dose. Dosimetry and dose-rate coverage (≥40 Gy/s) were evaluated for CTV Dmax, oral cavity Dmax and Dmean, mandible Dmax and D5cc, spinal cord Dmax, brainstem Dmax, chiasm Dmax, right and left optical nerve Dmax, right and left cochlea Dmean and Dmax, right and left parotid Dmean and Dmax, and lens Dmax. The setup and range uncertainties in the plan optimization were 3 mm and 3.5%.

## 3. Results

All eight patients (37.5% oropharynx, 25% oral cavity, 12.5% sinonasal, 12.5% nasopharynx, 12.5% salivary gland, all HPV negative) had experienced disease recurrence and were previously treated with conventional-rate IMPT “quad shot” RT (14.8 GyE in 4 3.7-GyE fractions, 2 per day), and were averaged to investigate the statistics for targets and OARs. The only variation was the dose rate (≥40 Gy/s for FLASH vs. 0.1 Gy/s for the conventional rate), minimum MU per spot (300–500 minimum MU per spot for FLASH vs. 1 minimum MU per spot for conventional rate), and fraction size (all patients were planned with 6 GyE vs. 10 GyE single-fraction plans). The 2D dose rate distribution for one representative patient and the accompanying DRVH comparison for a 6 GyE fraction is shown in Figure 1.

PBS Bragg peak FLASH demonstrated improved nominal OAR dose sparing via averaged metrics over PBS transmission FLASH on DVH analysis for 6-GyE and 10-GyE fractions, which was significant for some OARs and was more pronounced for 10 GyE than for 6 GyE (Table 1) (Figure 2). Conventional-rate IMPT showed OAR dose sparing similar to PBS Bragg peak FLASH and improved OAR dose sparing compared to PBS transmission FLASH via averaged metrics, with some dose comparisons showing significance (Table 1) (Figure 2). The OAR doses were close or equivalent to the PBS Bragg peak FLASH and conventional-rate IMPT plans via DVH (Table 1), except for the spinal cord Dmax, brainstem Dmax, left optic nerves, and right and left cochlea Dmax and Dmean, due to the plans chosen. The plan quality between 6 and 10 GyE was not different for conventional-rate IMPT. Therefore, 6 GyE was presented as a reference. All the FLASH plans were optimized for both 6 GyE and 10 GyE for balanced dose metrics and dose rate coverage.

All three modalities had acceptable CTV dose coverage, and both FLASH modalities had significantly increased CTV Dmax dose coverage relative to conventional-rate IMPT. Conventional-rate IMPT had a significantly more conformal CTV dose coverage compared to FLASH (Table 1) (Figure 2).

The PBS Bragg peak FLASH OAR FLASH dose-rate coverage was >90% (2-ms MST, 1200 MU). The DRVHs showed >90% V40 Gy/s dose-rate coverage across plans for both FLASH modalities, with similar CTV and OAR FLASH dose-rate coverage (Figure 3).

## 4. Discussion

PBS Bragg peak FLASH allows for conformal FLASH and dose sparing of tissue beyond the target via beam-specific range pull-back and compensation. PBS Bragg peak FLASH demonstrates similar dose plan quality to conventional-rate IMPT without accounting for FLASH normal tissue-sparing effects, as well as superior OAR dose sparing compared to PBS transmission FLASH. All the modalities provided effective CTV dose coverage, and both FLASH modalities generated sufficient > 40 Gy/s FLASH dose-rate coverage (>90%). The PBS delivery time was reduced by using the highest-energy (250 MeV) beams and eliminating energy switching. In-house spot map optimization allowed for multiple single-energy PBS Bragg peak FLASH beams to achieve conventional-rate IMPT-like dose plan quality while achieving ultra-high dose rates. These dosimetric advantages may translate to better clinical outcomes for salvage head and neck reirradiation.

This study’s excellent PBS Bragg peak FLASH CTV dose and dose-rate coverage, compared to conventional-rate CTV coverage using the same beam orientation and traversing the same tissue heterogeneity, suggests that PBS Bragg peak FLASH may demonstrate increased range uncertainty robustness in head and neck treatment planning. Future studies using this system can better quantify robustness in FLASH and conventional-rate head and neck plans using metrics such as beam uncertainty, fluence map optimization, spot-scanning sensitivity, heterogeneity, or group sparsity. Robust CTV dose and dose-rate coverage has been noted in prior studies of PBS Bragg peak FLASH in lung cancer treatment due to a relatively wide Bragg peak that minimizes perturbations [51,52]. More research is needed to verify if single-energy Bragg peak plans are more cost-effective by removing energy selection and only requiring range-shifter changes.

More dose organ sparing was observed at 10 GyE than at 6 GyE. The 6 GyE and 10 GyE conditions were compared to evaluate the OAR dose DVH and FLASH dose, as well as dose-rate coverage against the current biological evidence that the tissue-sparing effects of FLASH may be triggered at certain dose-rate and dose thresholds. The current study (Figure 2 and Figure 3) reflected research showing that larger dose thresholds reduce sub-FLASH regions, and V40 Gy/s FLASH dose-rate coverage increases to 90–100% above 5-Gy fractions [52]. In vitro studies have shown that FLASH tissue-sparing and cellular oxygen depletion starts at 40 Gy/s and 5–10 Gy [53], is most significant at 18 Gy, and there is no survival difference for doses < 5 Gy between FLASH and conventional-rate therapy [55]. Studies have indicated a lack of radiobiological FLASH effect at fraction sizes < 5 Gy, possibly related to radiosensitivity, oxygen tension, or FLASH reversibility in hypoxia, suggest that more research is needed to assess the temporal aspects of FLASH and its relationship to tissue oxygen content [48,55,56]. Smaller (<5 Gy) fraction sizes can complicate PBS-FLASH delivery through voxel-based dose-rate optimization, spot spacing, increased MU requirement, and decreased MST requirement [52]. The current study offers feasibility and proof-of-principle affirmations that there is likely a dosimetric advantage for higher fractional doses for the tumors, OARs, and the system used in the current study. This is clinically appealing in the setting of recurrent or radioresistant disease, where smaller (<2 Gy) fraction sizes with conventional-rate radiation have traditionally been used to reduce long-term toxicity. FLASH may shift this paradigm by enabling more aggressive treatment with higher fractional doses for radioresistant or recurrent disease. However, further work is needed to confirm a clinically meaningful FLASH effect and the biological advantages of higher fractional doses, given the inferior conformality of FLASH techniques.

Historically, salvage reirradiation has been technically feasible and has demonstrated improvements in disease-free survival, but no improvement in overall survival, and it is associated with a 10-year Grade 3+ toxicity of 40% [57]. Conventional-rate IMPT is often utilized in reirradiation to reduce toxicity [58], but it still carries toxicity risks [19,59]. PBS Bragg peak FLASH and its combination of FLASH dose rate and conformal dose sparing of OARs is appealing in this scenario to mitigate these risks, with the goal of reducing carotid injury [14,15,60] and TLI [16,61,62] and enabling more aggressive re-treatment dose-fractionation regimens that improve tumor control with similar or less toxicity.

Proton FLASH dose rates do not always entirely cover the OAR in Bragg peak or transmission methods [42], possibly due to sub-FLASH dose rates in low-dose regions. The impact on normal tissue sparing in regions of low dose and sub-FLASH dose rate requires further elucidation [53,55,63]. Conversely, the PBS Bragg peak itself may be wider than anticipated due to energy straggling and stochastic energy loss, which was observed in the current study’s increased FLASH CTV coverage [51]. Thus, PBS Bragg peak FLASH delivery using the highest energy may suffer from a lack of freedom, causing reduced target uniformity. Despite optimizing beams, range, patient positioning, MU per spot, and spot parameters [52,64], this technical limitation can be difficult to mitigate, as larger volumes, more fields, or lower MUs per spot require more spots and greater switching time to deliver the prescription dose, reducing the V40 Gy/s dose-rate coverage [44,52]. The radiobiological implications for toxicity and tumor control in overlapping FLASH and sub-FLASH dose rate regions and increased toxicity risk just beyond the Bragg peak [19,59], as well as how single-field, multi-field, or fractionated FLASH plans may result in differing biological effects [42], warrant future investigation. Theories regarding potentially increased RBE just beyond the distal Bragg peak in proton therapy and potential risk to PTV-adjacent OARs may also come into play, as transmission FLASH would avoid this hypothetical risk. PBS Bragg-peak FLASH may potentially increase this risk due to that greater degree of slowing and kinetic energy release that results from FLASH protons slowing to conform to the Bragg peak, and the true distal Bragg peak RBE of conventional-rate IMPT and PBS Bragg-peak FLASH has yet to be compared [19,42,59]. This information would greatly improve ongoing initiatives to develop patient-specific inverse-planning software that optimizes treatment delivery and dose through incorporating and weighting of the Bragg peak location, dose–volume deposition, dose rate, linear energy transfer, and OAR proximity [65].

This proof-of-principle study’s DVH and DRVH analyses were limited by small patient numbers. Another limitation and area for future improvement is the metrics used to describe FLASH delivery. DADR captures instantaneous dose rates, ignores dead times, and is considered a conservative estimate of dose-rate coverage, but it is unclear whether DADR adequately encapsulates FLASH’s mechanisms [23,66,67]. We introduced the V40 Gy/s dose-rate metric from multiple-dose thresholds and single-beam dose scenarios to denote OAR and CTV FLASH dose-rate coverage. As the FLASH effect for multibeam scenarios has not been fully characterized, a limitation of this method is that the dose rate is computed for single-beam scenarios. FLASH ultra-fast delivery should mitigate uncertainty from intra-beam motion, yet the current study did not account for uncertainty due to inter-beam motion or gantry rotation. New systems, beam-angle optimization, and hardware may be necessary to enable future FLASH delivery [42,68]. OAR dose thresholds in PBS-BPF can be challenging due to the zero exit dose. FLASH may alter planning goals, as OAR doses may have to be optimized to a FLASH threshold rather than to a dose constraint. There is ample opportunity for characterizing biological effects such as free-radical scavenging, oxidant load, and catalase activity in normal tissues and cancer at different FLASH dose rates [66]. As the FLASH delivery time is too short for reoxygenation, repopulation, or redistribution, more biological work must be done to elucidate changes in radiosensitivity, DNA damage, and DNA repair [23,24,66].

## 5. Conclusions

PBS Bragg peak FLASH can deliver conformal plans, ultra-high FLASH dose rates, and superior OAR dose sparing relative to PBS transmission FLASH. It also had comparable DVHs to conventional rate IMPT. All the modalities had excellent CTV coverage. For FLASH, OAR sparing was more pronounced at 10 GyE than at 6 GyE. There was >90% V40 Gy/s dose rate coverage across structures for both FLASH modalities.

## Figures and Tables

**Figure 1 cancers-15-03828-f001:**
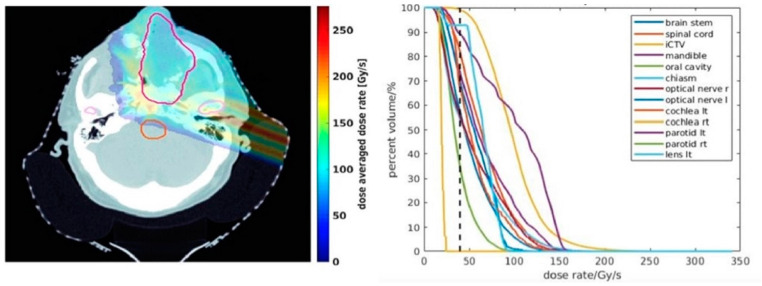
Dose-averaged dose rate (DADR) curves depicting volumetric dose rate coverage for a PBS Bragg peak FLASH plan using minimum 400 MU per spot, 2-ms MST, and 6 GyE per fraction.

**Figure 2 cancers-15-03828-f002:**
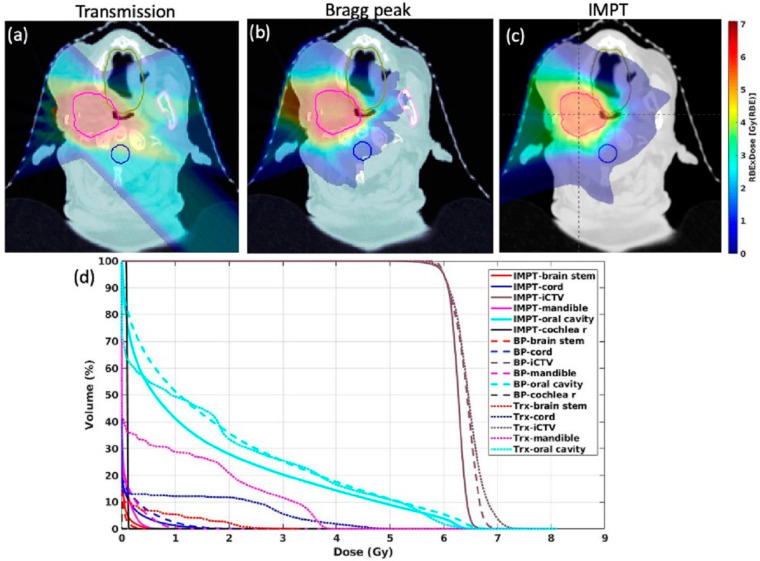
Dose–volume histogram (DVH) curves (**d**) depicting OAR and CTV dose coverage for an example patient for which (**a**) PBS transmission FLASH, (**b**) PBS Bragg peak FLASH, and (**c**) conventional-rate IMPT plans were created.

**Figure 3 cancers-15-03828-f003:**
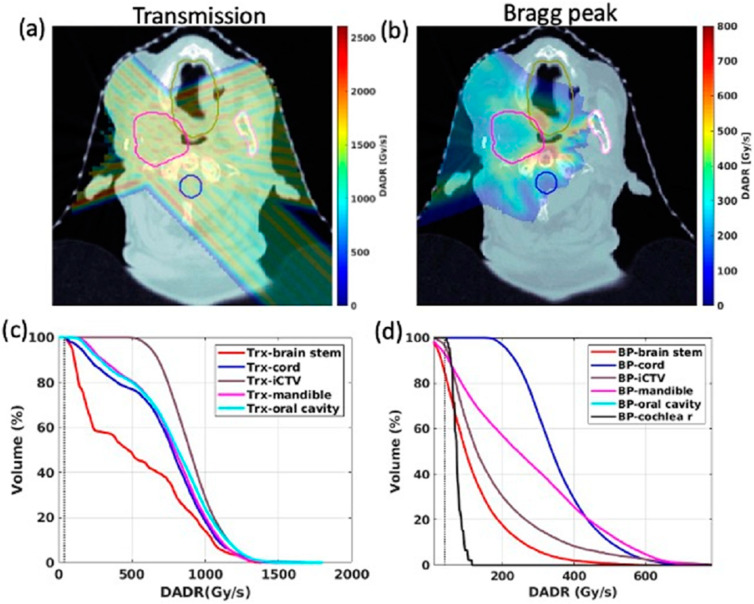
Dose–rate volume histogram (DRVH) curves depicting OARs and CTV dose-rate coverage for an example patient using (**a**,**c**) PBS transmission FLASH, and (**b**,**d**) PBS Bragg peak FLASH. The dashed vertical lines indicate the 40 Gy/s dose rate threshold.

**Table 1 cancers-15-03828-t001:** Dose metric comparisons between conventional-rate IMPT, PBS transmission FLASH, and PBS Bragg peak FLASH plans for 8 patients, with 6 GyE per fraction of conventional-rate IMPT presented as a reference. All doses were normalized to the prescription dose.

Dose Metrics	CONV-IMPT 6 Gy/Fraction (%)	6 GyE/Fraction					10 GyE/Fraction				
		TPF (%)	CONV-IMPT vs. TPF *p*-Value	BPF (%)	CONV-IMPT vs. BPF *p*-Value	TPF vs. BPF *p*-Value	TPF (%)	CONV-IMPT vs. TPF *p*-Value	BPF (%)	CONV-IMPT vs. BPF *p*-Value	TPF vs. BPF *p*-Value
CTV Dmax	109.5 ± 2.5	121.2 ± 4.7	**<0.001**	114.7 ± 3.8	**0.009**	**0.011**	118.5 ± 4.7	**0.002**	115.0 ± 6.0	**0.013**	0.124
Oral Cavity Dmax	90.9 ± 35.5	81.2 ± 50	0.227	76.8 ± 25.2	0.167	0.396	81.6 ± 49.2	0.226	78.6 ± 22.5	0.171	0.423
Oral Cavity Dmean	21.1 ± 14.5	27.5 ± 25.4	0.118	25.8 ± 22.6	0.145	0.266	28.3 ± 25.4	0.093	25.8 ± 22.5	0.144	0.206
Mandible Dmax	81.2 ± 35.4	101 ± 21	**0.020**	81.7 ± 33.5	0.471	**0.011**	104.6 ± 20	**0.022**	87.6 ± 27.0	0.243	**0.039**
Mandible D5cc	53.1 ± 38.4	68.4 ± 26.3	0.059	54.2 ± 41.3	0.382	0.078	72.8 ± 26	0.056	53.2 ± 36.3	0.492	**0.024**
Spinal Cord Dmax	34.6 ± 22.5	65.4 ± 17.7	**0.019**	37.3 ± 24.5	0.122	**0.025**	65.8 ± 16.9	**0.028**	35.4 ± 22.4	0.296	**0.032**
Brainstem Dmax	22.8 ± 35.3	34.2 ± 38.8	0.080	28.7 ± 39.9	0.086	0.254	35 ± 40	0.080	31.8 ± 46.8	0.160	0.341
Chiasm Dmax	18.3 ± 33.6	22.8 ± 36.8	0.360	20.8 ± 34.7	0.198	0.422	23.4 ± 38.2	0.352	20.4 ± 33.3	0.277	0.382
Optic Nerves RT Dmax	22.6 ± 39.2	25.9 ± 40.6	0.278	22.5 ± 36.8	0.479	0.173	27.9 ± 43.3	0.261	17.5 ± 32.2	0.087	0.163
Optic Nerves LT Dmax	21.6 ± 40.1	33.9 ± 52.7	0.151	27.5 ± 43.9	0.187	0.120	35.9 ± 55.6	0.159	28.6 ± 44.6	0.214	0.110
Cochlea L Dmax	19.5 ± 36.8	27.9 ± 45.1	0.122	22.6 ± 40.5	0.096	0.149	28.5 ± 45	0.147	20.2 ± 35.8	0.335	0.123
Cochlea L Dmean	15.4 ± 30.4	22.8 ± 36.4	0.147	17.2 ± 30.7	0.194	0.132	24.2 ± 37.7	0.168	14.2 ± 26.5	0.282	0.125
Cochlea R Dmax	10.3 ± 23.9	12.6 ± 19.6	0.387	9.6 ± 17.9	0.408	0.332	12.4 ± 19.1	0.403	9.0 ± 17.6	0.328	0.318
Cochlea R Dmean	6.3 ± 14.5	12.2 ± 18.9	0.194	7.4 ± 13.9	0.172	0.241	11.8 ± 18.3	0.205	7.0 ± 13.6	0.252	0.235
Parotid L Dmax	44.7 ± 46.1	63.7 ± 33.6	0.143	44.2 ± 40.6	0.464	0.134	62.3 ± 34.4	0.141	44.8 ± 41.9	0.492	0.148
Parotid L Dmean	6.3 ± 6.9	20.5 ± 14.2	0.051	11.1 ± 14.8	0.180	0.174	18.6 ± 11.9	**0.046**	7.4 ± 8.1	0.260	0.065
Parotid R Dmax	54.4 ± 52.8	55.3 ± 49.9	**0.050**	45.9 ± 52.5	0.324	0.051	56 ± 46.8	0.102	47.1 ± 49	0.271	0.058
Parotid R Dmean	9.3 ± 9.9	15.1 ± 16.2	**0.031**	14.9 ± 19.8	0.107	0.486	14.8 ± 15.9	**0.034**	8.9 ± 10.5	0.167	0.070
Lens LT Dmax	11.8 ± 20.3	19.1 ± 29.9	0.174	19.2 ± 31.8	0.089	0.490	25.4 ± 42	0.170	17.3 ± 31.4	0.108	0.291
Lens RT Dmax	8.8 ± 17.6	5.6 ± 13.6	0.196	10.5 ± 14.2	0.175	0.064	5.3 ± 13.1	0.196	7.3 ± 13.3	0.264	0.051

Bragg peak FLASH (BPF), conventional dose rate (CONV), intensity-modulated proton therapy (IMPT), pencil-beam scanning (PBS), transmission proton FLASH (TPF). bold font indicates statistical significance.

## Data Availability

The research data are stored in an institutional repository and will be shared upon request to the corresponding author.

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
