# Peer review of "Proton Bragg Peak FLASH Enables Organ Sparing and Ultra-High Dose-Rate Delivery: Proof of Principle in Recurrent Head and Neck Cancer"

_cancers, 2023, doi:10.3390/cancers15153828_

Round 1
Reviewer 1 Report
The purpose of this study is to assess the benefits of ultra-high dose rate and FLASH proton therapy in the treatment of recurrent head and neck cancer.
The subject of the paper is scientifically interesting and clinically meaningful. The work presented in this paper relies on planning tools previously developed by the authors and described in literature.
Material and methods:
When the 6-GyE single-fraction and 10-GyE single-fraction plans were created, how were the target dose prescriptions and the OAR dose constraints adapted with respect to the conventional protocol (14.8 GyE in 4 fractions) to take into account the different fractionation?
The minimum spot duration for the minimum spot MU was set at 2ms. What is the beam current at isocenter corresponding to this situation? Is the beam current assumed to be the same for all spots and equal to this current? If not, was there a maximum on the beam current?
Has the BP-FLASH solution been validated against measurements where the universal range shifter and a clinically realistic 3D range modulator were used in a similar way they are used in the paper? If not, this should be mentioned in the discussion.
The authors did not mention the use of apertures in PB-FLASH and they should comment about this aspect.
Were the BP-FLASH plans simulated with any limitation in field size or SOBP length with respect to the other two techniques?
What was the typical airgap between the end of the snout and the patient?
Did the conventional IMPT plan include a range shifter? How thick? What was the typical airgap?
Results
Table 1 suggests that the population of patients included in this study showed a significant variability in target volume and/or geometry, as the distribution of values around the mean seem quite broad. For instance, for both brainstem and chiasm the standard deviation was larger than the mean. It would be interesting to see the results also in a way that is not heavily impacted by patient-to-patient variation. The table should also include a metric for target coverage, such as CTV D98.
How were setup and range uncertainties considered in treatment plan optimization?
"PBS Bragg-peak FLASH had a lower low-dose volume than did conventional-rate IMPT but had a larger high-dose volume, extending into the parotid, brainstem, vertebrae and oropharynx (Figure 2)" Does this sentence mean that BP-FLASH achieved better sparing at low doses? If so, this seems counterintuitive to me. Also, unless I am missing something, figure 2 does not suggest that BP-FLASH is better than conventional IMPT at lower doses.
Discussion
"The robustness of this study’s PBS Bragg-peak FLASH CTV dose and dose-rate coverage reflects robust CTV dose and dose-rate coverage noted in prior studies due to a relatively wide Bragg peak that minimizes perturbations". Please provide some data about plan robustness in the 'Results' section. Given the significantly higher CTV Dmax of the two FLASH techniques, robustness of Dmax in OARs close to the CTV should be verified too.
"We anticipate that by including any FLASH-mediated normal-tissue sparing effect, biological effective dose and clinical outcomes will be superior to conventional-rate IMPT in future in vivo studies." It would be really good if we could already anticipate results of this kind, but I think that this sentence is premature given the current lack of data about the benefits of FLASH in human patients.
"The current study offers feasibility, proof-of-principle affirmation that there is likely a dosimetric advantage for higher fractional doses".
This is true provided that there is FLASH effect for this fraction size and these organs at risk, and that DADR is the correct way to calculate the dose rate. One can also see these results as an indication that a non-zero FLASH effects is needed for FLASH dose distributions not to be worse than conventional IMPT. In other words, part of the FLASH effect will be necessary to compensate for the inferiority of the FLASH techniques.
"Mini-beam arrays may hold the potential for high dose rates, focusing properties, non-conventional microstructures, and vascular changes [68,69]. As FLASH delivery time is too short for reoxygenation, repopulation, or redistribution, more biological work must be done to elucidate changes in radiosensitivity, DNA damage, and DNA repair [23,24,65,70,71]. " I find these final sentences a bit out of the context of this work.
Reviewer 2 Report
This study is very interesting, and important. Before publishing, I have two questions.
1. Could you show us the characteristics of 8 patients?
2. It is said that the RBE become higher in distal end. Could you show us your thoughts about the impact for high dose per fraction and FLASH?
English is well written.
